# Rare Hematologic Malignancies and Pre-Leukemic Entities in Children and Adolescents Young Adults

**DOI:** 10.3390/cancers16050997

**Published:** 2024-02-29

**Authors:** Amber Brown, Sandeep Batra

**Affiliations:** Division of Pediatric Hematology, Oncology and Stem Cell Transplant, Department of Pediatrics, Riley Hospital for Children, 705 Riley Hospital Drive, Indianapolis, IN 46202, USA; browambe@iu.edu

**Keywords:** rare leukemia, myelodysplastic neoplasm, myelodysplastic syndrome, lymphoproliferative disorders, predisposition syndromes, diagnosis, genomic, treatment, outcome

## Abstract

**Simple Summary:**

There are several rare types of hematologic malignancies and germline predispositions syndromes in children and adolescent young adults. In this descriptive review, we briefly describe rare hematologic malignancies, myelodysplastic neoplasms, and predispositions syndromes in children for which we have cohort outcome data and summarize emerging concepts in pathogenesis, diagnosis, prognostication, and treatment.

**Abstract:**

There are a variety of rare hematologic malignancies and germline predispositions syndromes that occur in children and adolescent young adults (AYAs). These entities are important to recognize, as an accurate diagnosis is essential for risk assessment, prognostication, and treatment. This descriptive review summarizes rare hematologic malignancies, myelodysplastic neoplasms, and germline predispositions syndromes that occur in children and AYAs. We discuss the unique biology, characteristic genomic aberrations, rare presentations, diagnostic challenges, novel treatments, and outcomes associated with these rare entities.

## 1. Introduction

Acute lymphoblastic leukemia (ALL) and acute myeloid leukemia (AML) are the predominant types of leukemia in children (age 0 to 14 years) and adolescent young adults (AYAs) (age 15 to 39 years). Myelodysplastic neoplasms, de novo or secondary to predisposition syndromes, carry a risk of progression to AML. Additionally, germline predisposition syndromes may progress to other myeloid neoplasms or ALL. In children and AYAs, there are several rare myelodysplastic neoplasms, AML, and predisposition syndromes that occur in <5% of cases, but are important to recognize due to their unique presentation, poor outcomes, and the potential for novel therapy. This descriptive review focuses on these rare entities (Figure 1), highlighting genomic and molecular features, unique presentations, diagnostic challenges, outcomes, and current treatment approaches, as well as potential novel therapies.

## 2. Myelodysplastic Neoplasms, Rare Entities

### 2.1. Myelodysplastic Neoplasms

Myelodysplastic neoplasms (abbreviated MDS by the World Health Organization (WHO)) are a heterogenous group of clonal hematologic disorders characterized by morphologic dysplasia of one or more cell lineages and ineffective hematopoiesis, leading to peripheral blood cytopenia and an increased but variable propensity for transformation into AML [1,2]. An overview of their molecular pathogenesis is summarized in Figure 2. The term myelodysplastic neoplasms replaced myelodysplastic syndromes in the WHO 2022 classification of hematopoietic and lymphoid tumors, underscoring their neoplastic nature [2].

MDS are rare in children, accounting for <5% of all hematologic neoplasms, and are extremely variable in terms of clinical features, disease progression, and outcomes [3,4]. The term “childhood MDS” defines a category of diseases biologically distinct from adult MDS [5]. Childhood MDS may be primary (de novo) or secondary, with the latter associated with predisposing conditions such as germline or somatic mutations, inherited bone marrow failure syndromes, or prior chemotherapy or radiation therapy [4]. The median age at diagnosis is 7–8 years, but MDS may present in children of any age with an approximately equal male-to-female ratio [4]. Presenting symptoms reflect persistent peripheral blood cytopenias and commonly include fatigue, bleeding, fever, and infections [4,6,7]. Hepatomegaly is usually absent; splenomegaly is rare; and lymphadenopathy is uncommon outside of concurrent infection [4,7]. Approximately 20% of children may be asymptomatic at presentation [7].

Diagnosing MDS is often challenging and requires a correlation of clinical features, peripheral blood values, bone marrow evaluation(s), cytogenetics, and molecular results [4,6]. The recommended threshold for the degree of morphologic dysplasia is 10% for each cell lineage, and the cytopenias are defined as hemoglobin < 13 g/dL (males) and <12 g/dL (females), absolute neutrophil count < 1.8 × 10^9^/L, and platelets < 150 × 10^9^/L [2]. Immunophenotyping via flow cytometry has shown variable success in diagnosing childhood MDS and may aid in approximately 10% of cases when immunophenotypic aberrations are present [8,9,10]. Immunohistochemistry may be helpful in confirming the diagnosis [4]. Cytogenetic and molecular testing have become increasingly important in the diagnosis and classification and include identifying predisposing somatic mutations and karyotype abnormalities and revealing a distinct mutational landscape of childhood MDS involving somatic mutations in *SETBP1*, *ASXL1*, *RUNX1*, and the RAS pathway [5,11].

The WHO classifies MDS as those having defining genetic abnormalities (“myelodysplastic neoplasms with defining genetic abnormalities”) or defining morphology (“myelodysplastic neoplasms morphologically defined”) [2] (Table 1). MDS is stratified based on the risk of leukemic transformation [6]. Disease-related risk and prognostication in adults are commonly assessed using the Revised International Prognostic Scoring System (IPSS-R), but this tool has limited utility in children and AYAs [12,13] (Table 2, Table 3 and Table 4). In general, children and AYAs with MDS without increased blasts (i.e., <5% blasts) have a better prognosis than those with increased blasts [4]. Additional poor prognostic features include monosomy 7 (associated with progression to more advanced MDS) [7], *SETBP1* mutations, and CD7 expression of myeloid blasts (both associated with decreased overall survival) [14,15].

Treatment of MDS in children and AYAs is challenging due to limited treatment options and the lack of consensus on optimal management. Treatment selection relies on the diagnosis, cytogenetic and molecular features, and clinical scenario. Current treatment strategies include observation (watch-and-wait), immunosuppressive therapy (IST), and allogeneic hematopoietic stem cell transplant (allo-HSCT) [4]. Allo-HSCT is the only curative treatment for childhood MDS, but due to associated morbidities, it is not offered as first-line therapy for all patients. It is routinely offered to patients with excess blasts, monosomy 7, del(7q), complex karyotype, transfusion-dependence, severe neutropenia, or MDS secondary to chemotherapy or radiation therapy [16]. Improved understanding of the molecular pathogenesis of childhood MDS and the identification of recurrent somatic driver mutations offer potential for the development of targeted therapies.

### 2.2. MDS with Low Blasts and 5q Deletion

#### 2.2.1. Pathologic and Cytogenetic Features

MDS with low blasts and 5q deletion occurs predominately in middle aged to elderly females [17], with fewer than 10 cases previously reported in children and AYAs [18,19,20,21,22,23,24,25]. It was first described by Van den Berghe et al. in 1974 [26], and further defined by Sokal et al. in 1975 [27]. The deleted region in the long arm of chromosome 5 contains genes important for hematopoiesis, leading to defective erythropoiesis and anemia [28,29]. The deletion can occur in isolation (5q syndrome) or in combination with other genetic abnormalities, the most common of which are mutations in *TP53* [30]. Isolated del(5q) is associated with a better prognosis, slower disease progression, a low risk of transformation to AML, and usually a favorable response to treatment [17,18,22,31,32].

#### 2.2.2. Treatment

Lenalidomide is an immunomodulatory agent that selectively suppresses the del(5q) clone [29]. Lenalidomide monotherapy has been evaluated in phase 2 and phase 3 trials in adults with low- and intermediate-1 risk del(5q) MDS, and outcomes have been compared to adults with non-del(5q) MDS with the same IPSS risk score [33,34,35]. In these trials, patients treated with lenalidomide had a significant erythroid response and a higher complete and partial cytogenetic response, with an improvement in median hemoglobin concentration resulting in a decreased transfusion requirement [33,34]. In adults with higher risk del(5q) MDS (i.e., intermediate-2 and high-risk per IPSS risk score), lenalidomide monotherapy resulted in only modest improvements in erythroid response and remission [36,37]. Improved results for adults with higher risk MDS were seen with lenalidomide and azacytidine combination therapy [38,39,40]. Contrary to isolated del(5q), adults with concomitant del(5q) and *TP53* mutations tend to have an inferior response to lenalidomide and an overall worse prognosis [37,41,42]. Despite promising results in adults, lenalidomide has not been investigated or utilized in the treatment of children and AYAs with del(5q) MDS.

Currently, there is no standard of care therapy for children and AYAs with del(5q) MDS. Treatment strategies have included observation [24], chemotherapy with steroids [21,25] or erythropoietin [20], and HSCT [18,19,22].

### 2.3. MDS with Low Blasts and SF3B1 Mutation

#### 2.3.1. Pathologic and Cytogenetic Features

Though well described in adult MDS, *SF3B1* mutations are rare in childhood MDS [4,5,14,43], with a solitary case report in a 17-year-old patient with Fanconi Anemia and MDS with ring sideroblasts and multilineage dysplasia [44]. *SF3B1* is the most common somatic spliceosome machinery gene mutation in adult MDS [4,45,46] and is associated with a highly homogenous phenotype characterized by erythroid dysplasia, ring sideroblasts, ineffective erythropoiesis, and often normal or nearly normal platelet and neutrophil counts [46,47,48]. Adults with *SF3B1*-mutated MDS are more often female, significantly older at the age of onset, and have a favorable outcome and lower risk of progression to AML [46,48,49,50,51]. In adults, the detection of *SF3B1* mutations is a favorable prognostic indicator associated with low IPSS-R categories and adult MDS with ring sideroblasts [46]. Due to the paucity of reported cases, the prognostic significance of *SF3B1*-mutated MDS in children and AYAs is unknown. In a cohort of 53 pediatric patients aged 11 months to 17 years with AML (52 with de novo AML, 1 with secondary AML), no recurrent *SF3B1* mutations were identified, suggesting that *SF3B1* mutations are unlikely to be a leukemogenic driver in childhood AML [52].

#### 2.3.2. Treatment

With the paucity of reported cases, there is no standard of care therapy for children and AYAs with MDS with low blasts and the *SF3B1* mutation. In in vitro and ex vivo studies in *SF3B1*-mutated AML, *SF3B1* modulators have shown potential as novel therapeutic targets in FLT3-internal tandem duplication (ITD) AML cells with a high allelic ratio and/or long ITD length [53]. Future studies are needed to elucidate the role of *SF3B1* modulators in the treatment of *SF3B1*-mutated MDS.

### 2.4. MDS with Biallelic TP53 Inactivation

#### 2.4.1. Pathologic and Cytogenetic Features

The *TP53* tumor suppressor gene functions as a negative regulator of cell proliferation. In MDS, *TP53* inactivation by mutation or deletion is associated with high-risk disease, rapid progression to AML, and poor prognosis in AYAs and adults [54]. *TP53* aberrations can be mono- or biallelic, each representing a subset of patients with distinct clinical phenotypes and outcomes [55]. Biallelic *TP53* inactivation results in loss of function of the *TP53* tumor suppressor gene and is a common and poor prognostic finding in adults with complex karyotypic MDS [54,55]. In general, *TP53* abnormalities are reported at low frequencies in childhood MDS [56,57,58], and biallelic *TP53* aberrations have not been described in the pathogenesis of childhood MDS [5,14]. However, biallelic *TP53* aberrations have been described in pediatric B-cell non-Hodgkin lymphoma and pleuropulmonary blastoma [59,60].

#### 2.4.2. Treatment

Due to the paucity of reported pediatric or AYA cases, there is no standard of care therapy for children and AYAs with MDS with biallelic *TP53* inactivation, but HSCT may be considered.

## 3. Myeloid Neoplasms, Rare Entities

### 3.1. AML

AML is relatively rare in children and AYAs, representing approximately 15% of acute leukemias. Despite maximally intensified chemotherapy and increased implementation of upfront HSCT for high-risk disease, the overall survival is in the 50 to 70% range, with approximately 40% of patients with de novo AML ultimately experiencing a relapse [61,62]. Large-scale, comprehensive genomic sequencing studies have improved the understanding of AML biology, risk stratification, prognostication, and treatment. AML was historically classified morphology using the French-American-British (FAB) classification (Table 5). AML is now risk-stratified by a combination of defining genetic features [2] (Table 6) and response to treatment. The diagnostic workup includes bone marrow morphology, flow cytometry immunophenotyping, cytogenetic studies (karyotyping and fluorescence in situ hybridization (FISH)), and comprehensive sequencing. Anthracyclines and cytarabine are key components of AML therapy for low- and intermediate-risk diseases, and consolidation with HSCT for high-risk diseases [61].

Efforts are ongoing to identify and develop effective targeted treatments, including small molecule inhibitors, cell surface antibody drug conjugates, and chimeric antigen receptor T cells (CAR-T), for use in upfront and relapsed/refractory settings [62,63,64,65,66,67,68]. Active targeted therapy trials in children and AYAs with AML are summarized in Table 7. (Table 7). The current Children’s Oncology Group AAML1831 trial (NCT04293562) has incorporated gilteritinib, a FLT3 inhibitor, in combination with standard chemotherapy and as maintenance therapy for all patients with FLT3 activating mutations or FLT3-internal tandem duplication (ITD) with a high allelic ratio (>0.1). The AAML1831 trial has also incorporated upfront gemtuzumab ozogamicin (CD33-targeted therapy) with conventional induction chemotherapy.

Briefly described below are subtypes of AML with rare genetic alterations. In general, treatment of these rare entities is per the current standard of care for AML therapy, with additions or exceptions noted below.

### 3.2. AML with MECOM Rearrangement

#### 3.2.1. Pathologic and Cytogenetic Features

The *MDS1* and *EVI1* complex locus protein (*MECOM*) gene (previously known as *EVI1*) is located on chromosome 3q26 and encodes a zinc-finger protein transcriptional regulator that is critical for hematopoiesis and hematopoietic stem cell self-renewal [69]. *MECOM* aberrations (i.e., overexpression, rearrangements, mutations) have been associated with poor prognosis of pediatric and adult myeloproliferative neoplasms [70,71,72,73,74,75,76,77,78,79,80]. *MECOM* rearrangements have been rarely reported in pediatric patients. The translocation of t(3;21)(q26;q22) results in the fusion of *MECOM* with *RUNX1* and has been reported in a pediatric patient with acute myelomonocytic leukemia [81], acute monoblastic leukemia [82], and secondary MDS/AML after treatment for T-cell ALL [83]. Additional *MECOM* fusions have been reported in pediatric patients, including *MECOM-RPN1* in a patient with AML with megakaryocytic differentiation [84] and *EIF4A2*-*MECOM* in a patient with severe congenital neutropenia and secondary AML [85]. Patients with these rare MECOM fusions ranged in age from 0 to 13 years.

In children and AYAs with AML, *MECOM* overexpression has been reported in up to 30% of patients [79,80,86,87]. Its frequency is significantly higher among younger patients (<10 years) with intermediate or high-risk AML, complex karyotypes, and *MLL* (11q23) abnormalities [79,80,86,87]. *MECOM* overexpression is an unfavorable prognostic factor that is associated with a lower event-free survival (EFS) (37.3–40% versus 50–68.4%) and overall survival (OS) (38.7–51% versus 68–78.9%), as well as a higher cumulative incidence of relapse [80,86].

#### 3.2.2. Treatment

Treatment strategies in children with AML with *MECOM* rearrangement have included chemotherapy and allo-HSCT [80,81]. With limited patient cases, there is no established standard of care. On AAML1831 (NCT04293562) (Table 7), *MECOM-RPN1* and *MECOM-RUNX1* fusions are classified as unfavorable prognostic markers, and AML patients harboring these fusions are treated per the high-risk arm (two induction cycles and one intensification chemotherapy cycle followed by allo-HSCT). All other *MECOM* aberrations are considered neutral prognostic markers, and the standard treatment consists of five cycles of chemotherapy (two induction cycles and three intensification cycles; Footnote 3, Table 7) if there is no evidence of residual AML (measurable residual disease (MRD) < 0.05%) at the end of induction 1, or three cycles of chemotherapy (two induction cycles and one intensification cycle) followed by allo-HSCT if there is evidence of residual AML (MRD ≥ 0.05%) at the end of induction 1 (Footnote 3, Table 7).

#### 3.2.3. MECOM-Associated Syndrome

*MECOM*-associated syndrome is a rare entity, reported in approximately 50 families, and is characterized by *MECOM* mutations, autosomal dominant amegakaryocytic thrombocytopenia, progressive bone marrow failure, pancytopenia, and radioulnar synostosis [88,89]. Additional forearm and hand anomalies, cardiac and renal anomalies, hearing loss, and/or B-cell deficiencies may also be present [89]. Due to congenital thrombocytopenia and progressive pancytopenia, allo-HSCT is often required in infancy [88,89]. Allo-HSCT with reduced-intensity conditioning has been shown to be both feasible and efficacious in infants with *MECOM*-associated syndrome [90]. One report linked *MECOM* mutations with hereditary hematologic malignancies, after two out of four family members with *MECOM* mutations and radioulnar synostosis developed MDS in adulthood [91].

### 3.3. AML with KAT6A-CREBBP Fusion

#### 3.3.1. Pathologic and Cytogenetic Features

The translocation t(8;16)(p11;p13) results in fusion of *KAT6A* (lysine acetyltransferase 6A, also known as the monocytic leukemic zinc-finger (*MOZ*)) to *CREB-binding protein* (*CREBBP* or *CBP*). Both KAT6A and CREBBP have histone acetyltransferase (HAT) activity and act as transcriptional co-activators [92]. In addition to its activity as a HAT, KAT6A is involved in the transcriptional regulation of several transcription factors with hematopoietic specificity [93]. *KAT6A* was first identified as a fusion partner of *CREBBP* in the context of AML, with the fusion (KAT6A-CREBBP) functioning as a transcriptional co-activator that promotes leukemogenesis [94,95,96]. An international retrospective study identified and characterized 62 children and AYAs age 0 to 18 years with *KAT6A-CREBBP* [97]. Patients were typically very young at diagnosis (median age 1.2 years; 50% of the cases were <2 years of age at diagnosis), often presenting within the first month of life [97]. In this cohort, outcomes were similar to other pediatric AML cases, outside of spontaneous remissions in a subset of young infants [97].

#### 3.3.2. Treatment

*KAT6A-CREBBP* fusion is considered an unfavorable prognostic marker (high-risk) on AAML1831 (NCT04293562), and the current standard of care would include two induction chemotherapy cycles and one intensification cycle, followed by a consolidative allo-HSCT [97]. Several targeted therapies are currently being developed that specifically inhibit KAT6 [98].

### 3.4. AML with DEK-NUP214 Fusion

#### 3.4.1. Pathologic and Cytogenetic Features

Translocation t(6;9)(p22;q34) results in the *DEK-NUP214* fusion gene [99]. This DEK-NUP214 fusion protein is a nucleoporin with altered nuclear protein transport. Increased expression of the DEK-NUP214 fusion protein is specific to myeloid lineage cells, underscoring its association with AML, though its role in myeloid leukemogenesis is not completely understood [100]. *DEK-NUP214* fusion has been reported in both AML and MDS in children and AYAs [101,102,103,104,105,106]. A report from the Children’s Oncology Group (2839 total patients, age range 0 to 29.8 years, treated on six consecutive AML trials) identified only 48 cases (1.7%) of the *DEK-NUP214* fusion [107]. *DEK-NUP214* fusion was associated with an older age (median age 12.6 years versus 8.9 years, *p* < 0.001), compared to *DEK-NUP214* negative AML patients, and the FAB morphology subtype M2 (*p* = 0.03) [107]. Patients with *DEK-NUP214* positive AML had outcomes similar to those of patients with AML with poor prognostic features, such as –7 or 5/del5q. Compared to patients with *DEK-NUP214* negative AML, patients with *DEK-NUP214* positive AML had a lower complete remission (CR) (67% versus 79%, *p* = 0.04), lower OS (39% versus 57%, *p* = 0.03), and higher relapse rates (64% versus 42%, *p* = 0.04) [107]. Both *DEK-NUP214*-positive AML and MDS are also highly associated with a co-occurring FLT3-ITD mutation [102,105,106,108,109]. In the Children’s Oncology Group cohort, however, outcomes in the *DEK-NUP214* positive AML group were independent of FLT3-ITD, with no significant difference in the OS between the FLT3-ITD-positive and -negative groups (40% and 27%, respectively, *p* > 0.9) [107]. A high risk of relapse and poor overall survival have also been reported in international cohorts of children and AYAs with AML [106].

#### 3.4.2. Treatment

*DEK-NUP214* is considered an unfavorable prognostic marker, and patients harboring this mutation should be treated per high-risk AML protocols (two induction cycles and one intensification cycle followed by allo-HSCT) (Footnote 3, Table 7).

### 3.5. AML with FLT3-ALM

#### 3.5.1. Pathologic and Cytogenetic Features

Somatic mutations of *FLT3* are among the most common mutations in AML, with a prevalence of 10 to 20% in children and AYAs with AML, including acute promyelocytic leukemia (APL) [110,111,112,113,114]. *FLT3* mutations include internal tandem duplication (ITD; FLT3-ITD) and missense mutations in the activation loop of the tyrosine kinase domain (TKD; FLT3-ALM or FLT3-TKD), with the former being more common [108]. These genetic aberrations result in the constitutive activation of the FLT3 receptor, resulting in increased cellular proliferation and leukemogenesis [115,116]. In a large cohort of 630 pediatric patients age < 21 years with de novo AML, FLT3-ITD was detected in 12% of patients (age 0.6–19.8 years) and FLT-ALM in 6.7% (age 0.3–19.7 years) [117]. In this cohort of children and AYAs, the progression-free survival (PFS) and OS of patients with FLT3-ITD were inferior to those of FLT3-wild type (WT) (31% versus 55%, *p* < 0.001), but the PFS and OS were similar in patients with FLT3-ALM compared to FLT3-WT (51% versus 55%, respectively, *p* = 0.8). A report from the Children’s Oncology Group of 104 pediatric patients age < 21 years with APL identified FLT3-ALM in 14% of the patients. In this cohort, patients with FLT3-ALM and FLT3-ITD had similar outcomes, including similar rates of induction death, complete remission, EFS, and OS [114].

#### 3.5.2. Treatment

Multiple FLT3 inhibitors have been developed for use in *FLT3*-mutated AML, classified by receptor binding and the mutation(s) they are active against. The multi-center Children’s Oncology Group AAML1031 trial investigated the feasibility and efficacy of incorporating sorafenib (a multi-kinase tyrosine kinase inhibitor (TKI)) when combined with standard AML chemotherapy in children and AYAs (age < 30 years) with de novo AML and FLT3-ITD with a high allelic ratio (>0.4) [118]. Compared to a nonrandomized control group, patients in the sorafenib-treated cohort had significantly improved EFS (55.9% versus 31.9%, *p* = 0.001) and lower relapse risk (17.6% versus 44.1%, *p* = 0.012) [118]. AML1031 also demonstrated that post-transplant sorafenib maintenance therapy for one year is feasible [119]. Midostaurin and gilteritinib are also TKIs used in FLT3-ITD adult patients, with the latter targeted more specifically against FLT3. Studies in adults utilizing midostaurin and gilteritinib, including small cohorts of patients with FLT3-ALM, have demonstrated improved treatment responses [120,121]. The RATIFY trial (NCT00651261) investigated the addition of midostaurin to chemotherapy in newly diagnosed FLT3-mutated adult patients. A sub-analysis of patients with FLT3-ALM, including patients aged 19.3 to 59.9 years (median age 48.8 years), showed longer EFS in the midostaurin-treated patients (45.2% versus 30.1%, *p* = 0.044, compared to the placebo arm). There were no further sub-analyses comparing AYAs and adult patients in this study [122]. Currently, the AAML1831 trial is investigating gilteritinib in combination with standard chemotherapy and 1-year maintenance chemotherapy after allo-HSCT or after completion of induction and intensification chemotherapy in children and AYAs age < 22 years with FLT3 activating mutations or FLT3-ITD with a high allelic ratio (>0.1) (Footnote 3, Table 7).

Recently, FLT3 has been targeted with CAR-T therapy. Preclinical in vitro data demonstrate the cytotoxic effects of targeted CAR-T cells on AML cell lines and primary AML cells, as well as a survival benefit in mouse xenograft models [67].

### 3.6. AML with Megakaryoblastic Differentiation, RBM15-MKL1 Fusion

#### 3.6.1. Pathologic and Cytogenetic Features

*RBM15-MKL1* (previously known as *OTT/MAL*) translocation t(1;22)(p13;q13) fuses the RNA-binding motif protein (*RBM15*) gene on chromosome 1 with the megakaryoblastic leukemia-1 (*MKL1*) gene on chromosome 22. *RBM15-MKL1* is a recurrent genetic aberration unique to non-Down syndrome acute megakaryoblastic leukemia (non-DS AMKL), occurring in up to 13% of cases [123,124,125,126]. AMKL is an AML subtype in which cell morphology resembles abnormal megakaryoblasts. Case reports of a *TERT* promotor variant in a young child with *RBM15-MKL1*-positive non-DS AMKL [127], a 3-way variant translocation (t(1;7;22)(p13;q21;q13)) in a neonate with non-DS AMKL [128], and a 4-way variant translocation (t(1;22;17;18)(p13;q13;q22;q12)) in an infant with non-DS AMKL [129] have been published. AMKL is rare in adults but accounts for 4–15% of newly diagnosed childhood AML [130]. Patients typically present at age < 2 years with anemia, thrombocytopenia, and organomegaly [131]. Clinical outcomes of *RBM15-MKL1*-positive non-DS AMKL in children and AYAs vary among studies. Inaba et al. and Schweitzer et al. describe intermediate event-free survival (EFS) and overall survival (OS) of 38–50% and 56–63%, respectively [124,132], whereas better outcomes (59% EFS, 70% OS) have been described by de Rooij et al. [125]. In these studies, patients were all < 18 years old, and the investigators speculated that differences in supportive care might have biased the outcome [124,125].

#### 3.6.2. Treatment

*RBM15-MKL1* fusion is considered a neutral prognostic marker on AAML1831, and the standard of care would include chemotherapy with or without allo-HSCT, pending response to induction chemotherapy (Footnote 3, Table 7). There are no known specific novel inhibitors targeting *RBM15-MKL1* specifically.

### 3.7. AML with Megakaryoblastic Differentiation, CBFA2T3-GLIS2 Fusion

#### 3.7.1. Pathologic and Cytogenetic Features

*CBFA2T3-GLIS2* fusion characterizes an AML subtype exclusive to pediatrics, accounting for approximately 5% of all pediatric AML and primarily found in non-DS AMKL in infants and AML in early childhood, with most patients age < 5 years [130,133,134]. The presence of the *CBFA2T3-GLIS2* fusion is associated with aggressive disease and a dismal prognosis, with OS ranging from 15 to 30% and EFS of 38% [84,125,133,134,135]. The CBFA2T3-GLIS2 fusion protein promotes leukemogenesis by upregulating transcription factors that ultimately result in enhanced self-renewal and inhibition of cell differentiation [133]. The fusion results from the cryptic inversion of chromosome 16, which is often missed in morphology and cytogenetic studies. Thus, diagnosis relies on a high CD56 expression, dim/negative CD45, CD38, and HLA-DR expression (termed RAM phenotype) [84,136]. Concomitant cytogenetic abnormalities are rare in patients with *CBFA2T3-GLIS2* fusion AML [135,137].

#### 3.7.2. Treatment

Given their poor prognosis, *CBFA2T3-GLIS2* fusion-positive patients are allocated to high-risk arms of pediatric treatment protocols, including AAML1831 (NCT04293562), and are candidates for allo-HSCT if negative MRD and complete remission are achieved. The *CBFA2T3-GLIS2* fusion is an attractive option for targeted therapies. In pre-clinical studies, CD56-directed antibody-drug conjugates effectively targeted *CBFA2T3-GLIS2* fusion-positive AML blasts [137]. Additionally, Le et al. demonstrated the efficacy of CAR-T directed against cell surface folate receptor alpha in vitro and in mouse xenograft models [68].

### 3.8. AML with RUNX1-CBFA2T3 Fusion

#### 3.8.1. Pathologic and Cytogenetic Features

*RUNX1-CBFA2T3* fuses *RUNX1* on chromosome 16 and *CBFA2T3* on chromosome 21 (t(16;21)(q24;q22)) and is a rare translocation described in de novo and therapy-related AML. In pediatric patients, it has been primarily described in children and AYAs aged 1 to 39 years [138,139,140,141,142,143,144,145]. Many patients with *RUNx1-CBFA2T3* had M1, M2, or M4 FAB subtypes, and a subset of these patients were found to have eosinophilia [140,141,142,144,146], but accumulation and evaluation of additional cases are needed to determine if these patients represent a clinically significant distinct subgroup. In an international cohort of 23 children and AYAs (median age 6.8 years, age range 1 to 17 years) with *RUNX1-CBFA2T3*, the fusion was associated with an overall favorable outcome with EFS and OS of 77% and 81%, respectively, and a 0% 4-year cumulative incidence of relapse [138].

#### 3.8.2. Treatment

Therapy has included chemotherapy with or without consolidative allo-HSCT. *RUNX1-CBFA2T3* fusion is considered a neutral prognostic marker, and the standard of care would include chemotherapy with or without allo-HSCT, depending on the response to induction chemotherapy. Given favorable outcomes, consideration should be given to risk-stratifying *RUNX1-CBFA2T3* to standard-risk AML therapy.

### 3.9. Acute Erythroid Leukemia

#### 3.9.1. Pathologic and Cytogenetic Features

Acute erythroid leukemia (AEL) is a rare subtype of AML characterized by the proliferation of immature erythroblasts in the bone marrow and peripheral blood. AEL is more common in adults but rarely occurs in children and AYAs, accounting for <5% of all AML cases [147,148]. The diagnosis of AEL is based on the presence of at least 50% erythroid precursors in the bone marrow and/or peripheral blood, along with evidence of dysplasia of ≥10% of the cells in two or more hematopoietic lineages. The genetic basis of AEL remains poorly defined, but both adult and pediatric cases show enrichment of *NUP98* rearrangements, as well as a variety of other somatic mutations [149,150,151,152]. The Children’s Oncology Group reported recurrent *NUP98* rearrangements in approximately 32% of pediatric AEL cases [149]. AEL is associated with a poor prognosis, with OS and EFS each accounting for approximately 20% of pure erythroid leukemia [149].

#### 3.9.2. Treatment

Given the poor prognosis of this entity, HSCT is preferred.

### 3.10. Myeloid Neoplasms with Eosinophilia and Defining Gene Rearrangement

#### 3.10.1. Pathologic and Cytogenetic Features

This rare group of myeloid malignancies was previously known as chronic eosinophilic leukemia (CEL) or idiopathic hypereosinophilic syndrome (HES). They are characterized by a clonal proliferation of myeloid and eosinophilic cells and are associated with gene fusions involving *PDGRA*, *PDGFRB*, *FGFR1*, or a protein tyrosine kinase such as *JAK2* (e.g., *PCM-JAK2*), resulting in constitutively active tyrosine kinase and proliferation of the abnormal clone [153]. Clinicopathologic presentation is heterogenous. Leukocytosis with eosinophilia, anemia, hepatosplenomegaly, lymphadenopathy, and skin lesions (rashes, ulcers) have been reported in children aged 0 to 14 years [153].

#### 3.10.2. Treatment

Treatment often involves a tyrosine kinase inhibitor (TKI) with or without chemotherapy and/or HSCT [153].

## 4. Syndromes Predisposing to Myelodysplastic and Myeloid Neoplasms, Rare Entities

### 4.1. Syndromes Predisposing to Myelodysplastic and Myeloid Neoplasms

A variety of syndromes with germline mutations may predispose to MDS and/or myeloid leukemias. Broadly, these may be classified as germline predisposition syndromes with or without pre-existing platelet disorders or a risk of organ dysfunction (Table 8). Constitutional pathogenic variants in *DDX41*, *ETV6*, *CEBPA*, *RUNX1*, *ANKRD26*, and *GATA2* are especially prone to an increased risk of developing hematologic malignancies [154]. The clinical presentation of the syndromes is variable but may include isolated cytopenias or pancytopenia, with or without bone marrow failure [155]. Accurate diagnosis of inherited predisposition syndromes prior to myelodysplastic or leukemic transformation is paramount for management, treatment, and family genetic counseling [155]. Allo-HSCT is often considered for a cure. International best practice consensus guidelines detailing recommendations on the HSCT timeline, genomic assessment, donor selection, and genetic counseling have been published [154]. Briefly described below are the rare entities among these syndromes.

### 4.2. Myeloid Neoplasms with Germline Predisposition without a Pre-Existing Platelet Disorder or Risk of Organ Dysfunction with CEBPA and DDX41 Mutations

These rare disorders are characterized by either *CEBPA* or *DDX41* mutations and are often inherited in families without a pre-existing platelet disorder or organ dysfunction [156,157,158,159]. Patients with familial AML with germline *CEBPA* mutations are typically younger at presentation than patients with de novo disease, with a median age of 25 years and without antecedent MDS or cytopenias [158]. However, age at onset is variable, occurring at 1.8 to 62 years [157,158]. Both *CEBPA* and *DDX41* mutations are associated with an increased risk of developing AML and MDS (*DDX41* mutations) [158,159,160,161].

### 4.3. Myeloid Neoplasms with Germline Predisposition and Pre-Existing Platelet Disorders with RUNX1, ANKRD26, and ETV6 Mutations

These germline mutations are associated with thrombocytopenia as well as an increased risk of developing MDS or AML. *RUNX1* mutations are associated with familial platelet disorder (FPD) with a predisposition to AML [157,161,162,163,164], while *ANKRD26* and *ETV6* mutations are associated with inherited thrombocytopenia and a predisposition to developing MDS and AML [162,165,166]. *RUNX1*-mutated families are heterogenous in mutation profile and phenotype [164]. In general, patients often have thrombocytopenia (platelet counts range from 70–145 × 10^9^/L) and associated mucosal bleeding, as well as eczema or psoriasis [161]. Childhood onset (age < 18 years) malignancy is observed in 50% of families with *RUNX1* mutations, although the median age (29 years) of onset is in young adulthood [161]. Treatment includes chemotherapy and allo-HSCT.

Patients with *ANKRD26*-related thrombocytopenia also present at variable ages, ranging from age 10 to 75 years with thrombocytopenia (platelet counts range from 8.5–85 × 10^9^/L), mucosal bleeding, and menorrhagia; life-threatening hemorrhages are rare [165]. In addition to AML and MDS, germline *ANKRD26* mutations have been less often associated with chronic myelogenous leukemia and chronic lymphocytic leukemia [165].

Patients with germline *ETV6* mutations similarly present with thrombocytopenia and mucosal bleeding, though some patients also have anemia and/or neutropenia at presentation [166]. In addition to MDS and AML, patients with germline *ETV6* mutations have developed other hematologic malignancies (i.e., pre-B cell ALL, mixed-phenotype acute leukemia, chronic myelomonocytic leukemia, multiple myeloma), as well as skin and colorectal cancer [166].

### 4.4. Myeloid Neoplasms with Germline Predisposition and Risk of Organ Dysfunctions

Children and AYAs with genetic-based syndromes including, but not limited to, inherited bone marrow failure syndromes (BMFS), telomere biology disorders (e.g., dyskeratosis congenita), neurofibromatosis type 1 (NF1), Noonan syndrome, or Noonan syndrome-like disorders, and Trisomy 21 have an increased predisposition to developing myeloid neoplasms [162]. These syndromes may affect several organ systems as well as hematopoiesis, underscoring their variable phenotypic presentation and progression to malignancy. Inherited BMFS may predispose to MDS and AML, whereas RASopathies such as NF1 and Noonan/Noonan-like syndromes may predispose to juvenile myelomonocytic leukemia (JMML). Spontaneous or germline mutations in the *GATA2* gene may disrupt normal hematopoiesis and are associated with the development of myeloid neoplasia, including AML, MDS, and chronic myelomonocytic leukemia [161,167].

## 5. B-Cell Lymphoid Proliferations, Rare Entities

### 5.1. B-Cell Lymphoid Neoplasms

ALL is the most common pediatric malignancy [168]. B-cell ALL accounts for 75–80% of cases [168] and commonly presents with cytopenias and associated symptoms due to bone marrow involvement by leukemic blasts, as well as constitutional symptoms such as fever. Extramedullary involvement of the central nervous system (CNS) or testis in males may also occur. Similar to AML, diagnostic workup includes bone marrow morphology, flow cytometry, immunophenotyping, cytogenetic studies (karyotyping and FISH), and comprehensive sequencing in high-risk patients. Risk stratification and prognostication include age, white blood cell count (WBC), genomic aberrations, and response to induction chemotherapy [169]. WHO classification emphasizes the critical role of genomic aberrations in risk stratification and prognostication in ALL [170].

Highlighted below are rare precursor and mature B-cell ALL subtypes. Currently open Children’s Oncology Group trials, AALL1731 (standard risk, patient age > 1 to ≤31 years) (NCT03914625) and AALL1732 (high risk, patient age > 1 to ≤25 years) (NCT03959085) are incorporating targeted therapies: blinatumomab (bi-specific anti-CD19 and anti-CD3 fusion protein) in AALL1731 and inotuzumab ozogamicin (anti-CD22 antibody) in AALL1732, respectively. Both blinatumomab and inotuzumab ozogamicin have been used in the relapsed/refractory setting [171,172]. There is increasing interest in investigating the use of CD19 CAR-T earlier in treatment for very high risk patients, patients at high risk of relapse, and patients in their first relapse [173,174].

### 5.2. Near Haploid ALL (24 to 30 Chromosomes)

#### 5.2.1. Pathologic and Cytogenetic Features

Near-haploid ALL is a rare subtype of B-cell ALL that accounts for approximately 2–4% of cases and is usually associated with a loss or absence of multiple chromosomes, most commonly chromosomes X, Y, 6, 7, 8, 9, 17, 18, and 20 [175,176]. The role of chromosome loss in leukemogenesis has not been elucidated. Near-haploidy presents at younger ages (median age 5 years, range 1–19 years) [176]. Both near-haploidy and low-hypodiploid groups present with approximately equal incidence in males and females and with a relatively low WBC of <50 × 10^9^/L [176]. Near-haploid ALL has been shown to harbor a distinct mutational profile as compared to low-hypodiploidy [177,178]. Greater than 70% of near-haploidy cases harbor activating receptor tyrosine kinase (RTK) and RAS singling alterations, with additional common somatic alterations involving histone modifiers, NF1, CREBBP, *CDKN2A/B*, *IKZF3*, and *PAG1* [175,177]. Near-haploidy is a poor prognostic marker (EFS approximately 28%, OS approximately 34%) associated with induction failure and an increased risk of relapse [178,179].

#### 5.2.2. Treatment

Due to poor outcomes, treatment includes intensified chemotherapy and HSCT [176,179]. In relapse or refractory settings, CD19 and CD22-targeting immunotherapy and anti-CD19 CAR-T therapy have been employed [180]. Holmfeldt et al. investigated the sensitivity of hypodiploid ALL cell lines and xenografts to MEK, PI3K, and PI3K/mTOR inhibitors [177]. Both PI3K and PI3K/mTOR inhibitors decreased tumor proliferation, making PI3K pathway inhibition a promising potential treatment option [177].

### 5.3. Low Hypodiploid ALL (31 to 39 Chromosomes)

#### 5.3.1. Pathologic and Cytogenetic Features

Low hypodiploid is also a rare subtype of B-cell ALL, accounting for <3% of all cases [175], and is usually associated with retention of disomies X/Y, 1, 5, 6, 8, 10, 11, 14, 18, 19, 21, and 22 [176]. As with near haploidy, the selective advantage of chromosome loss in leukemogenesis is poorly understood. Low hypodiploidy occurs at all ages, but with a higher median age (age 15 years), and the incidence increases with age [176,178]. As outlined above, its gender incidence and the presenting WBC at diagnosis are similar to those of near-haploid ALL, but its genomic landscape is distinct [175,176,177]. Genomic alterations in *TP53* are common and a hallmark of low hypodiploid ALL, with greater than 90% of cases harboring loss-of-function mutations in *TP53* [181]. Importantly, *TP53* alterations in low hypodiploid ALL have been detected in nontumor cells in almost half of pediatric cases, suggesting an association with Li-Fraumeni syndrome [177]. Additional genomic alterations include *RB1* and *IKZF2* [177,182]. Outcomes are poor, with EFS and OS approximately 37% and 40%, respectively [179].

#### 5.3.2. Treatment

Similar to ALL with near-haploidy, treatment includes intensified chemotherapy and HSCT. CAR-T may also be considered in the relapsed/refractory setting and in patients with relapsed/refractory B-cell ALL harboring *TP53* alterations. Interestingly, decitabine may improve CAR-T cell efficacy [183]. Hypodiploid ALL clones have aberrant RAS and PI3K signaling, and PI3K and PI3K/mTOR inhibitors may be an intriguing novel treatment option [177]. Given the high frequency of *TP53* mutations and their presence in nontumor cells, patients should be evaluated for germline *TP53* mutations and Li-Fraumeni syndrome and, if identified, offered genetic counseling [176,177].

### 5.4. Mature B-Cell ALL

#### 5.4.1. Pathologic and Cytogenetic Features

Mature B-cell ALL is a rare subtype of B-cell ALL characterized by the presence of mature-appearing B-cells with surface IgM with light-chain restriction in the absence of immature B-cell antigens and FAB-L3 morphology [184]. It is often associated with t(8;14) or its variants and is molecularly characterized by MYC aberrations, including MYC overexpression, rearrangements, and fusion gene products (C-MYC/IGL or IGK), which result in leukemogenesis [184]. Based on these features, mature B-cell ALL is often considered the leukemic phase of Burkitt lymphoma. Patients are predominately male, with a median age of 9 years at presentation, and tend to have favorable outcomes (OS greater than 80%) [185].

Rare cases of mature B-cell ALL with *KMT2A* rearrangement have been reported in children and AYAs age 0 to 24 years, with the vast majority being age < 2 years and presenting with extramedullary involvement of the liver, spleen, skin, kidney, and occasionally the central nervous system [184,186]. Given the small numbers and varying treatment approaches, outcomes cannot be generalized.

#### 5.4.2. Treatment

Cases of mature B-cell ALL have been treated with standard lymphoma protocols [184,186]. Case reports of children, AYAs, and adults aged 2 to 78 years using CAR-T therapy for relapsed Burkitt lymphoma have been published [187].

### 5.5. B-Lymphoblastic Leukemia/Lymphoma

#### 5.5.1. Pathologic and Cytogenetic Features

B-lymphoblastic leukemia/lymphoma (LBL/L) is an aggressive subtype of leukemia/lymphoma that can be associated with specific genetic abnormalities, such as a translocation between the *TCF3* gene on chromosome 19 and the *PBX1* gene on chromosome X or a translocation of the *IGH* gene on chromosome 14 and *IL3*, resulting in the fusion gene product *TCF3-PBX1* or *IGH-IL3*, respectively. In early studies, B-LBL/L with *TCF3-PBX1* showed a poor response to chemotherapy. However, with more recent chemotherapy, it has shown an improved prognosis, albeit with an increased risk of CNS relapse [169,188].

#### 5.5.2. Treatment

Treatment typically consists of chemotherapy [188].

### 5.6. Chronic Lymphocytic Leukemia

#### 5.6.1. Pathologic and Cytogenetic Features

Chronic lymphocytic leukemia (CLL) is a lymphoproliferative disorder characterized by the accumulation of monoclonal B cells in the blood, bone marrow, or lymphoid tissues [189]. It predominantly occurs in older adults (median age 72 years at diagnosis) and is very uncommon in children and AYAs, with fewer than 10 previously reported cases [189,190]. The presentation of CLL in childhood may be associated with organomegaly or lymphadenopathy, as opposed to hyperleukocytosis (as in adults) [190]. CLL may be associated with specific genetic abnormalities such as Trisomy 12 or TP53 gene mutations [189], which may contribute to leukemogenesis. The outcomes of CLL in children are favorable, with reported CR rates of 44% and overall response rates of 95% [191].

#### 5.6.2. Treatment

Standard treatment consists of chemoimmunotherapy with fludarabine, cyclophosphamide, and rituximab [189,191]. In the relapsed/refractory setting, CD19 CAR-T therapy has been demonstrated to be efficacious in adults [192,193].

## 6. T-Cell Lymphoid Proliferations, Rare Entities

### 6.1. T-Cell Large Granular Lymphocytic Leukemia

#### 6.1.1. Pathologic and Cytogenetic Features

T-cell large granular lymphocytic leukemia (T-LGL) is a rare, classically clinically indolent disease of adults, with fewer than 10 cases reported in children and AYAs (age range 2 to 13 years) [194,195,196,197,198]. LGL is a mature, peripheral T-cell neoplasm that develops due to clonal proliferation of cytotoxic (CD8+) T cells that infiltrate organs such as bone marrow, liver, and spleen, resulting in cytopenias. LGL is also associated with dysregulation of or abnormalities involving the immune system and presents with coexisting autoimmune phenomena, such as rheumatoid arthritis, autoimmune neutropenia, or thrombocytopenia [196,197,198,199,200]. It has also rarely been described in adult solid organ transplant recipients [201].

Antigenic and cytokine stimulation of multiple molecular pathways has been implicated in the pathogenesis of T-LGL, including JAK/STAT, PI3K/AKT, FAS/FAS-L, RAS, NF-KB, and the sphingolipid rheostat pathway [202]. Lymphocytes in T-LGL may be difficult to morphologically distinguish from non-neoplastic cytotoxic lymphocytes; thus, diagnosis relies heavily on molecular techniques such as polymerase chain reaction (PCR) utilizing a probe for T-cell receptor (TCR)γ for diagnosis and flow cytometry for confirmatory immunophenotyping [200].

#### 6.1.2. Treatment

Treatment is indicated for patients with symptomatic anemia, neutropenia with recurrent infections, and/or autoimmune conditions [203]. Immunosuppressive therapy (commonly methotrexate, cyclosporine, and cyclophosphamide) often mitigates sequelae but may or may not result in cure [194,203]. In addition to immunosuppressive therapy, chemotherapy or HSCT may also be utilized [194,200,204,205,206]. In the relapsed/refractory setting, allo-HSCT successfully induced remission in a 13-year-old, suggesting allo-HSCT be considered for a definitive cure in children [194]. Molecular pathways implicated in T-LGL pathogenesis offer potential for targeted therapeutic approaches [203], though none have been studied in children and AYAs.

## 7. Conclusions

Described here are rare hematologic malignancies, myelodysplastic neoplasms, and predisposition syndromes in children and AYAs occurring with a low (<5%) frequency. Collectively, they are a heterogenous group, each of which often has a unique and complex genetic basis and clinical phenotype. Recognition and accurate diagnosis of these entities are important and may direct treatment choices and impact outcomes. Specific molecular lesions, genomic alterations, or unique immunophenotypes may help identify novel treatment approaches in these patients. Treatment usually involves conventional chemotherapy, immunotherapy, CAR-T therapy, HSCT, molecular targeted therapies, or a combination of the above. There is often a lack of consensus on optimal management, so the treatment may need to be individualized. Despite recent advances in classification schema, diagnostic methods, and treatment landscape, there is still much to learn about these rare hematologic malignancies and pre-leukemic entities.

## Figures and Tables

**Figure 1 cancers-16-00997-f001:**
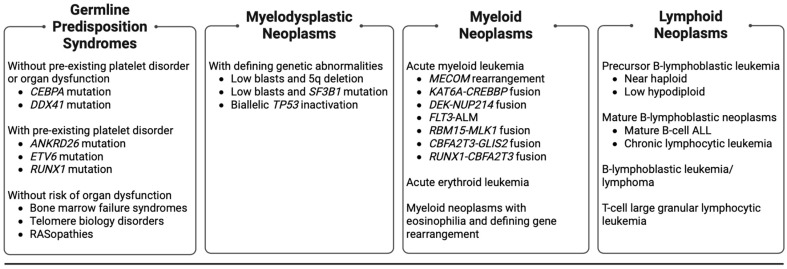
Rare (occurring in <5% of cases in children and AYAs) hematologic malignancies, myelodysplastic neoplasms, and germline predisposition syndromes in children and AYAs included for review. AYAs: adolescent young adults. ALL: acute lymphoblastic leukemia.

**Figure 2 cancers-16-00997-f002:**
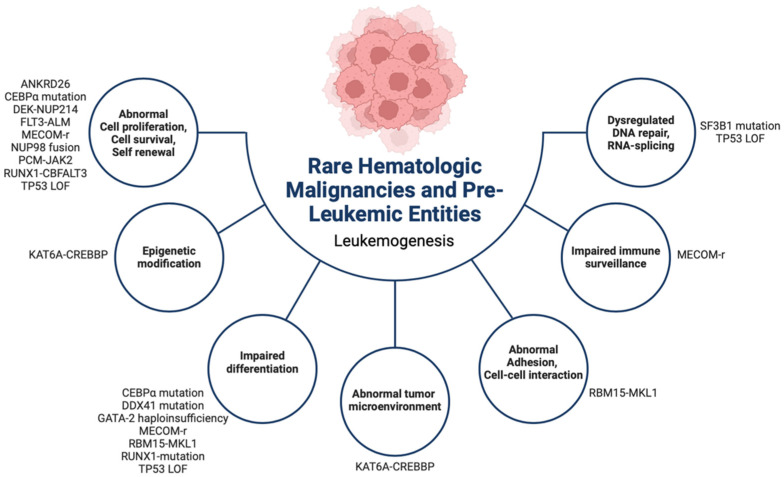
Summary of the leukemogenesis of rare (occurring in <5% of cases in children and AYAs) hematologic malignancies, myelodysplastic neoplasms, and germline predisposition syndromes in children and AYAs. AYAs: adolescent young adults. r: rearrangement, LOF: loss of function.

**Table 1 cancers-16-00997-t001:** WHO 2022 classification of myelodysplastic neoplasms (modified from Khoury et al. [2]).

MDS Subtypes	Blasts	Cytogenetics	Mutations
**MDS with defining genetic abnormalities**			
MDS with low blasts and 5q deletion	<5% BM and <2% PB	5q deletion with or without 1 other abnormality other than 7q deletion or monosomy 7	
MDS with low blasts and *SF3B1* mutation	<5% BM and <2% PB	Absence of 5q deletion, monosomy 7, or abnormal karyotype	*SF3B1*
MDS with biallelic *TP53* inactivation ^1^	<20% BM and PB	Often complex	*TP53* ^1^
**MDS morphologically defined**			
MDS with low blasts	<5% BM and <2% PB		
MDS with increased blasts ^2^	5–19% BM or 2–20% PB		
MDS, hypoplastic ^3^			

^1^ Two or more *TP53* mutations or 1 *TP53* mutation with an additional *TP53* alteration (i.e., copy number loss or copy neutral loss of heterozygosity). ^2^ MDS with increased blasts (MDS-IB) is sub-divided into three groups, MDS-IB1, MDS-IB2, and MDS with fibrosis, based on blast percentage. MDS-IB2 may also be defined by presence of Auer rods. ^3^ Less than or equal to 25% bone marrow cellularity. WHO: World Health Organization. MDS: myelodysplastic neoplasm, BM: bone marrow, PB: peripheral blood, SF3B1: splicing factor 3B subunit 1, TP53: tumor protein 53 gene.

**Table 2 cancers-16-00997-t002:** Revised International Scoring System (IPSS-R) for evaluating cytogenetic risk in adult MDS (modified from Greenberg et al. [12] and Marques et al. [6]).

Risk Category	Cytogenetic Aberration
Very good	–Y, del(11q)
Good	Normal, del(5q), del(12p), del(20q), double including del(5q)
Intermediate	del(7q), +8, +19, isochromosome(17q), Any other single or double independent clones
Poor	–7, inv(3)/t(3q)/del(3q), double including –7/del(7q), Complex karyotype: 3 aberrations
Very poor	Complex karyotype: >3 aberrations

MDS: myelodysplastic syndrome. del: deletion, inv: inversion, t: translocation.

**Table 3 cancers-16-00997-t003:** Revised International Scoring System (IPSS-R) adult MDS prognostic scoring system (modified from Greenberg et al. [12] and Marques et al. [6]).

Variable	Score						
	0	0.5	1	1.5	2	3	4
Cytogenetics	Very good	-	Good	-	Intermediate	Poor	Very poor
BM blast (%)	≤2	-	>2–<5	-	5–10	>10	-
Hemoglobin (g/dL)	≥10	-	8–<10	<8	-	-	-
Platelets (×10^9^/L)	≥100	50–<100	<50	-	-	-	-
ANC (×10^9^/L)	≥0.8	<0.8	-	-	-	-	-

MDS: myelodysplastic syndrome. BM: bone marrow, ANC: absolute neutrophil count, - indicates not applicable.

**Table 4 cancers-16-00997-t004:** Revised International Scoring System (IPSS-R) adult MDS risk stratification (modified from Greenberg et al. [12] and Marques et al. [6]).

IPSS-R Risk Category	Cumulative Score
Very low	≤1.5
Low	>1.5–≤3.0
Intermediate	>3.0–≤4.5
High	>4.5–≤6.0
Very high	>6.0

MDS: myelodysplastic syndrome.

**Table 5 cancers-16-00997-t005:** French-American-British classification of acute myeloid leukemia.

FAB Subtype	Description
M0	AML without differentiation
M1	AML with minimal differentiation
M2	AML with differentiation
M3	APL, hypergranular
M3v	APL, microgranular
M4	AMML
M4Eo	AMML with eosinophilia
M5	Acute monocytic leukemia
M6	Acute erythroblastic leukemia
M7	Acute megakaryoblastic leukemia

AML: acute myeloid leukemia. FAB: French-American-British. APL: Acute promyelocytic leukemia. AMML: Acute myelomonocytic leukemia.

**Table 6 cancers-16-00997-t006:** WHO 2022 classification of acute myeloid leukemia.

**AML with defining genetic abnormalities ^1^**
APL with *PML-RARA* fusion
AML with *RUNX1-RUNX1T1* fusion
AML with *CBFB-MYH11* fusion
AML with *DEK-NUP214* fusion
AML with *RBM15-MRTFA* fusion
AML with *BCR-ABL1* fusion
AML with *KMT2A* rearrangement
AML with *MECOM* rearrangement
AML with *NUP98* rearrangement
AML with *NPM1* mutation
AML with *CEBPA* mutation
AML, myelodysplasia-related
AML with other defined genetic alterations
**AML defined by differentiation ^2^**
AML with minimal differentiation
AML without maturation
AML with maturation
Acute basophilic leukemia
Acute myelomonocytic leukemia
Acute monocytic leukemia
Acute erythroid leukemia
Acute megakaryoblastic leukemia

WHO: World Health Organization. APL: Acute promyelocytic leukemia. AML: acute myeloid leukemia. ^1^ 20% bone marrow blast requirement for AML subtypes with defining genetic abnormalities was eliminated in WHO 2022 classification, with the exception of *BCR-ABL1* fusion and *CEBPA* mutation. ^2^ 20% bone marrow blasts required for diagnosis.

**Table 7 cancers-16-00997-t007:** Genomic and cell surface targets in pediatric AML currently under clinical study.

Target	Genomic Alteration ^1^	Agent	Clinical Trial ^2^
FLT3	FLT3-ITD	Gilteritinib Quizartinib	NCT04293562 ^3^ NCT04240002 NCT03793478
	FLT3-TKD	Gilteritinib	NCT04293562 ^3^
BCL2	-	Venetoclax	NCT03194932 NCT03826992 NCT03236857 NCT04898894 NCT05317403
IDH	*IDH2*	Enasidenib	NCT04203316
Menin	*KMT2Ar*, *NPM1*, *NUP98*	SNDX-5613	NCT04065399 NCT05326516
E-selectin	E-selectin ligand expression	Uproleselan	NCT05146739
XPO1	-	Selinexor	NCT04898894
CD33	-	Gemtuzumab ozogamicin (ADC) Anti-CD33 CAR-T	NCT02724163 NCT03971799
CD123	-	Anti-CD123 CAR-T	NCT04318678 NCT04678336

AML: acute myeloid leukemia. ITD: internal tandem duplication. TKD: tyrosine kinase domain. ADC: antibody drug conjugate. CAR-T: chimeric antigen receptor T cells. ^1^ indicates not applicable. ^2^ National Clinical Trials Network identifier provided. Includes trials for upfront and relapsed/refractory settings. ^3^ Children’s Oncology Group trial AAML1831.

**Table 8 cancers-16-00997-t008:** Syndromes with germline mutations predisposing to myelodysplastic and/or myeloid neoplasms.

**Without pre-existing platelet disorder or organ dysfunction**
*CEBPA* (CEBPA-associated familial AML) *DDX41* * *TP53* (Li-Fraumeni syndrome) *
**With pre-existing platelet disorder**
*RUNX1* (familial platelet disorder with associated myeloid malignancy) * *ANKRD26* (thrombocytopenia 2) * *ETV6* (thrombocytopenia 5) *
**With risk of organ dysfunction**
*GATA2* (*GATA2*-deficiency) *SAMD9* (MIRAGE syndrome) *SAMD9L* (*SAMD9L*-related ataxia pancytopenia syndrome) Biallelic germline *BLM* (Bloom syndrome) Inherited bone marrow failure syndromes Fanconi anemia Shwachman-Diamond Syndrome Severe congenital neutropenia Telomere biology disorders RASopathies Neurofibromatosis type 1 Noonan syndrome or Noonan syndrome-like disorders Casitas B-lineage lymphoma syndrome Trisomy 21 (Down syndrome) *

* Lymphoid neoplasms may also occur.

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
