# Peer review of "Rare Hematologic Malignancies and Pre-Leukemic Entities in Children and Adolescents Young Adults"

_cancers, 2024, doi:10.3390/cancers16050997_

Round 1
Reviewer 1 Report (Previous Reviewer 3)
Comments and Suggestions for Authors
The authors fully reply to all queries.
Comments on the Quality of English LanguageEnglish language is satisfactory
Author Response
Dear Reviewer,
Thank you for reviewing our manuscript and for your help to enhance it.
A revised version is attached
best,
Sandeep Batra and Amber Brown
Reviewer 2 Report (New Reviewer)
Comments and Suggestions for Authors
After carefully reviewing the manuscript, comments and suggestions of two other reviewers, as well as author's reply to reviewers, I find that the authors have addressed all concerns raised by the reviewers, and the manuscript has improved significantly. Thus, I have nothing to add to comments of other reviewers. The manuscript is comprehensive and can be used as a guide to whose working/studying in a field of genetics/molecular diagnostics of rare hematologic malignancies and pre-leukemic etities in Children and AYA.
Author Response
Dear Reviwer
thank you for your comments and review
The revised and final version is attached,
best wishes,
Sandeep Batra and Amber Brown
This manuscript is a resubmission of an earlier submission. The following is a list of the peer review reports and author responses from that submission.
Round 1
Reviewer 1 Report
Comments and Suggestions for Authors
Dear authors,
thank you for your manuscript. I have a few questions/ remarks.
1. Methodology: it is named as review. But what kind of review? How did you choose and select the rare leukemias/ neoplasms? Why did you include both children and AYA - over this age range we have so many biologically diffent leukemias. And you did not define the age range of AYA and how in your literature search you did include them.
2. The title is leukemias, but then you talk about MDS and lymphomas? Why?
3. Introduction: focused on pedicatric only. What about the AYA population you are mentioning in your title?
4. diseases: you mainly present data from children and adults, but the AYA population is mostly missing.
5. results/ discussion/ conclusion: you are describing a few selected liquide neoplasms - some extensive, other very short. What is your format presenting each disease? And should not available clinical trials - especially for the AYA group - be discussed? Are all entities analyzed separately for the pediatric and AYA age group.
Reviewer 2 Report
Comments and Suggestions for Authors
The review provides valuable insights and a comprehensive overview of the current state of knowledge regarding rare leukemias in children and AYAs. It underscores the challenges in this field and the need for continued research and collaboration to improve diagnosis and treatment strategies.
A few suggestions for authors:
An introductory table summarizing the different leukemia types, their incidence, genetic markers, and treatment specifics could be a quick reference for readers.
Prioritize an in-depth exploration of pediatric-specific leukemias, dedicating the majority of the text to these diseases. Following this, briefly address adult-type Myelodysplastic Syndromes (MDS), highlighting their rarity in children and key differences from pediatric conditions. Ensure that the proportional allocation of text clearly reflects the emphasis on pediatric diseases, with adult MDS serving more as a comparative reference.
Concluding with a stronger emphasis on future research directions and identifying gaps in current knowledge would be beneficial. This could guide upcoming research efforts and highlight areas where international collaborative efforts are needed.
Reviewer 3 Report
Comments and Suggestions for Authors
Brown and Batra’s text reports a review of rare leukemia in pediatric patients and young adults.
The authors describe the genetic characteristics and other considerations, in particular the outcome, of some very rare forms of leukemia in the population in question.
the part relating to myelodystopian syndromes is particularly complete and updated. Tables 1 to 4 are very significant and also useful for non-field readers.
The section on myeloid and lymphoid leukemia could benefit from summary tables with epidemiological, biological (genetic) characteristics and the main therapeutic indications currently available. Authors could also report a figure (similar to MDS) summarizing the most frequently observed genetic damage.
In the case of B-leukemia and B-lymphomas, the authors should also report the increasing use of CAR-T technology.
Reviewer 4 Report
Comments and Suggestions for Authors
The manuscript by Brown et al. is a review of rare leukemia in the pediatric population. The authors aim to focus on “leukemias” that account for less than 1% in frequency.
Many things could be improved in the current manuscript, most related to the poorly defined focus, which precludes a more comprehensive analysis of specific entities that may arise in pediatric populations.
Specifically:
1. Is the focus of the manuscript acute precursor myeloid/lymphoid neoplasms? Or any hematological neoplasm with a leukemic presentation? The authors have included in the review article entities classified as mature B-cell lymphomas, such as chronic lymphocytic leukemia, which is a mature neoplasm and stands out from all the other precursor neoplasms discussed and should be a separate section. Similarly, the authors have included T-cell large granular lymphocytic leukemia, a mature T-cell lymphoproliferative neoplasm. Please see point #2.
2. Myelodysplastic syndromes are not leukemic diseases. Their inclusion in this review needs to be more explicit. Unless the authors are focusing on precursor myeloid neoplasms? If this is the case, mature B or T- lymphoproliferative disorders should be excluded (point #1).
3. The entities included are not discussed in depth, and this is because the focus of the manuscript is not well-defined. For example, although pediatric myelodysplastic syndromes constitute rare occurrences, only the most frequent causes are reviewed (not those with less than 1% frequency, as stated in the introduction). Those associated with rare germline mutations that lead to rare cases of familial MDS/AML are not discussed. Similarly, the pathology description is poor; flow cytometry testing may aid in only 10% of the cases (when immunophenotypic aberrancies are present), and immunohistochemistry is rarely used to distinguish hematogones vs. myeloblasts (page 2, line 61). There needs to be a detailed description of the morphological findings identified on bone marrow biopsies that establish the diagnosis of myelodysplastic syndromes. It would be more helpful if each section (such as pediatric MDS) were divided into sub-sections, such as pathological findings, laboratory findings, management, etc.
4. Otherwise, if the authors intend to focus on entities that account for less than 1% in frequency, an in-depth description of those specific entities will be required, rather than a general description of myelodysplastic syndromes and/or acute myeloid leukemia with very few lines on one or two of low-frequency entities.
5. Once the focus is defined, the introduction section must be expanded accordingly.